# Handwritten Multi-Scale Chinese Character Detector with Blended Region Attention Features and Light-Weighted Learning

**DOI:** 10.3390/s23042305

**Published:** 2023-02-18

**Authors:** Manar Alnaasan, Sungho Kim

**Affiliations:** Department of Electronics Engineering, Yeungnam University, 280 Daehak-ro, Gyeongsan-si 38541, Republic of Korea

**Keywords:** handwritten Chinese character detection, blended region attention features, light-weighted learning

## Abstract

Character-level detection in historical manuscripts is one of the challenging and valuable tasks in the computer vision field, related directly and effectively to the recognition task. Most of the existing techniques, though promising, seem not powerful and insufficiently accurate to locate characters precisely. In this paper, we present a novel algorithm called free-candidate multiscale Chinese character detection FC-MSCCD, which is based on lateral and fusion connections between multiple feature layers, to successfully predict Chinese characters of different sizes more accurately in old documents. Moreover, cheap training is exploited using cheaper parameters by incorporating a free-candidate detection technique. A bottom-up architecture with connections and concatenations between various dimension feature maps is employed to attain high-quality information that satisfies the positioning criteria of characters, and the implementation of a proposal-free algorithm presents a computation-friendly model. Owing to a lack of handwritten Chinese character datasets from old documents, experiments on newly collected benchmark train and validate FC-MSCCD to show that the proposed detection approach outperforms roughly all other SOTA detection algorithms

## 1. Introduction

Handwritten Chinese character detection in historical documents has attracted more attention in the computer vision field due to its different applications, such as image retrieval, document analysis, and text translation. Chinese manuscripts hold valuable recordings. Therefore, many efforts have been made, ranging from traditional machine learning to recent deep learning techniques, to preserve and translate these manuscripts. However, challenges due to complexity in the character layout, ruined records, the density of characters in the documents, and the huge diversity in character scales make the translation problem difficult and laborious.

Recently, character-level detection, with its breakneck progress in the deep learning branch, has been handled as a feature extraction problem performed by a convolutional neural network CNN. In this regard, hierarchical, sequence-based, and segmentation-based models [1,2,3] have been presented to compensate for the lack of datasets, and pre- and post-processing approaches [4] provide pretty good solutions for precise detection tasks. However, it is worth mentioning that these methods may suffer from over-segmentation errors, and they might inaccurately position characters in a document. In the matter of handling the noise level of the image resulting from character segmentation and detection, a modified single shot multibox detector is implemented [5]. It provides strong detection on scale. However, using multi prior boxes is not an efficient method. 

Generally speaking, existing methods, often constructed with various feature map structures to handle multi-size characters, cannot yield high-quality character positions. An open issue is obtaining more information on localization. In addition, expensive learning makes the task of detection much slower.

In this paper, we investigate a light, simple, and dynamic detector that precisely predicts multi-scale Chinese characters in valuable old documents. Using multi-stage feature maps blended gradually with a candidate-free proposal, it adapts to multi-size characters to achieve a light-weight network.

The contributions of this work are as follows:A light-weighted feature stacking-based network is presented to simply, rapidly, and precisely predict multi-size Chinese characters in old documents. By stacking feature maps gradually, we achieve accurate predictions. Furthermore, we are freed of anchor candidates by looking for the center of the character and then regressing to the four corners of the bounding box, which reduces time-wasting issues and makes for simpler training.Due to the lack of datasets with Chinese characters in historical documents, we worked collaboratively with a team from Kyungpook National University to collect and analyze a new dataset containing tough challenges that were then used for training and testing our model.The proposed algorithm provided great results that beat state-of-the-art methods in terms of accuracy and efficiency.

Section 2 presents some works that are relevant to our paper subject. Section 3 demonstrates our novel method in detail. Section 3 evaluates the performance of the proposed algorithm compared with previous related methods. We conclude and summarize our work in Section 5.

## 2. Related Work

Text different-level detection in old documents Peng [6] studied the fully convolutional network for segmentation and recognition of Chinese text in an end-to-end manner. Another end-to-end style was applied by Peng [7] himself for page-level recognition of handwritten Chinese text using weakly supervised learning. Ma [8] analyzed joint layout detection and recognition for historical document digitization of old characters. However, all previously mentioned methods ignored the challenging detection at the character level, which we considered in this work.Chinese character-level detection in old documents Wang et al. [3] proposed a weakly supervised learning method in a character classifier for over-segmentation-based handwritten Chinese text recognition. Using a two-stage convolutional network, each line is over-segmented into a sequence of parts that are integrated to produce the character candidates. Afterward, a recognition score is set for each character class to produce a result for a recognized string. Aleskerova and Zhuravlev [1] used a hierarchical classifier of two-stage to solve the problem of a high number of Chinese character classes. First, all classes of similar features are grouped into one cluster and trained by the first-stage network to determine the number of groups. Then, the errors obtained are corrected by the second-stage classifier in order to assign correct labels to corresponding classes. Zhu et al. [2] took advantage of over-segmentation and a CRNN with an attention-based method to investigate one-to-many attention problems over character recognition output.Feature extraction with a feature fusion model Ronneberger et al. [9] was the first to invent the feature fusion model, called U-Net, to localize the area of abnormality in biomedical images. After that, many attempts [10,11] were presented to achieve better localization and multi-scale detection.Liu et al. [12] improved the detection of Oracle characters by embedding feature fusion at different levels based on ResNet101 as the backbone feature extractor. Zheng et al. [13] received strings as an input sequence and then attached the features of each character and word level to extract local features of various sizes. Finally, using a deep pyramid structure, they can capture global features. Yuan et al. [14] added a so-called ‘Gate’ after each feature map before uniting it to extract powerful features and remove any existing noise. Such features captured by a gate-based layer are more effective.

## 3. Methodology

In this section, we introduce the proposed feature-fusion-based algorithm with a skip connection for more precise detection of the character area. The main idea is to predict multi-scale characters with cheap parameters. In order to achieve this, we consider a higher level of feature extraction to find the center of each character, applying a Gaussian filter to increase the clarity of positive center points.

### 3.1. Network Architecture

Characters that range from large to small make consideration of a model with features from multi-levels in high demand because fine information can be obtained from earlier layers in a CNN, while enclosing coarser information means we need later layers. To remedy this, and by adopting U-like design [15], we stack feature maps gradually. Afterward, to reach ultimate accuracy in finding the correct character region, one more residual branch provides blend connections, inspired by the ResNet50 structure [16], to blend a previous map with the current one. This adds valuable details about character locations by improving the extraction of features. In addition, a style of bottle-neck is attained for the up-sampling branch to make our model advantageous at handling the issue of character detection with cheap parameters.

A backbone is obtained as a dense-free convolutional layer, based on ResNet-52 [16], for Chinese character-level detection in historical documents. That backbone yields final score maps of multiple channels, as follows: two channels for backgrounod and foreground classification, one channel for the center point, and four channels to regress to the four corners of the bounding box. Our system is viewed as a schematic in Figure 1. Two branches are employed: down-scaling as feature abstraction and up-scaling as feature stacking. The down-scaling branch is the trunk, which can be a network pretrained on an ImageNet [17] dataset with convolutional and pooling layers. For feature extraction, five levels of activation maps are used from the trunk with different sizes (0.25, 0.125, 0.0625, and 0.03125) of the input image. A stack branch uses feature maps from the trunk branch to aggregate features level-by-level, and to concatenate features from the prior layer after up-sampling each level. To avoid optimization issues due to a very deep network, ReLU and batch-normalization are exploited after each unpooling layer in the stack branch and after each unpooling in the blend connection. Furthermore, after each stack block, ReLU and batch-normalization are added. Suppose *N* is the operation using ReLU and batch-normalization, then the whole operation will be defined as:(1)$$S_{i}=\{\begin{array}{ll} N\left(Conv_{3\times 3}\left(Conv_{1\times 1}\left(Cat\left(unpool\left(S_{i+1}\right),d_{i}\right)\right)\right)\right) & for\;i=4,\;3,\;2\\ \;\;unpool(Conv_{3\times 3}\left(S_{i+1}\right)) & for\;i=1\\ \;\;cat\left(d_{5},\;d_{5}/2\right) & for\;i=5 \end{array}$$
(2)$$B_{i}=\{Cat\left(S_{i},\;N\left(unpool\left(S_{i+1}\right)\right)\right)\;\;\;\;\;\;\;\;\;\;\;\;\;\;\;\;\;\;\;\;\;\;\;\;\;\;\;\;\;\;\;\;\;\;\;\;\;\;\;\;\;\;\;for\;\;\;\;i=4,3,2\;\;\;\;\;\;\;\;\;\;\;\;\;$$
where $S_{i}$ represents the stack branch, and $B_{i}$ the blend connectio. Cat(·) indicates concatenation along the feature dimension. In the stack branch, each feature map that comes from each trunk’s stage after pooling layers is concatenated with the present feature map after doubling its size through unpooling, and then, a 1×1 Conv layer is used to lower the number of channels and lessen the computations. Afterward, inserting a 3×3 Conv layer to stack the feature information produces the fused output. To make the final output for the second, third, and fourth convolutional layers, an additional blend connection that concatenates the current output with the previous unpooling output is obtained to enhance information about the location of the character and to boost detection accuracy. However, the first output for the first stack layer is obtained by fusing the middle and last feature maps of the last convolutional layer in the trunk branch. Finally, the endmost output is produced by 3×3 Conv layer after unpooling, as in [18], and before feeding to the last destination, which is 1×1 Conv layer. After attaching two siblings 1×1 Conv layers, the head prediction is appended. For simplicity, it consists of two channels for classification, one channel for center-ness, and four channels for bounding box regression.

### 3.2. Bounding Box Generation

Similar to other applications that use heatmaps because of their adaptability to treat the problem of inaccurately bounded regions in ground truth data as a starting point to detect key points in response to fully convolutional layers [18,19,20], our algorithm goes further to find the center of each character and draw a bounding box around it automatically. Heatmaps are the same size as stacked feature maps. One is a heatmap with three channels indicating the center point, background, and foreground; the other four channels are in another heatmap to regress to the four box boundaries from the point of center. 

The multi-scale default boxes algorithm regresses the bounding box of the target based on these anchors after considering the center of the anchors as the location on the input image. Our detector directly classifies and regresses the bounding box of the target at the location during the training phase, which is carried out with a Fully Convolutional Network (FCN), as in [21]. Figure 2 explains the process of finding the center point for each character with reference to the bounding box label. We consider the point of each center to be positive; negative is assigned to the other location. After that, we automatically regress to the four corners of the target-bounding box.

### 3.3. Loss Function Head

Our model is implemented with two sub-tasks: classification and regression. Classification provides a feature map with a 3D vector for each location, indicating the background, foreground, and center scores. Regression provides a feature map with a 4D vector for the four distances from the character’s center to the four corners of the bounding box. However, before sending the last stacked feature map to the detection head, we append a single 3×3 Conv layer to reduce the number of channels to 256. After that, to formulate the final center heatmap and bounding box map, two siblings 1×1 Conv layers are attached. 

For background and foreground classification, we use cross-entropy loss to obtain the score of location (i,j) as 0 or 1, which means that it is located in the background or foreground, respectively. This loss function is donated as $L_{cl}$.

The predicted heatmaps are sized (Hn×Wn), where H and W are the height and width of the heatmap, and n is the down-sampling factor. Unlike the semantic segmentation algorithm, which labels each pixel of an image separately, we assign positive to the center point of the character, with a Gaussian heatmap for each character. This Gaussian distribution is used to learn the center score at each character’s location (i,j) to alleviate the confusion of the negative examples surrounding the positive examples, and to ease the learning phase. A full explanation of the process of the 2D Gaussian filter is shown in Figure 3. In theory, its formula is:(3)$$G_{ij}\left(x,y,\sigma_{w},\sigma_{h}\right)=e^{-\left({\left(i-x\right)}^{2}/2\sigma_{w}^{2}+{\left(j-y\right)}^{2}/2\sigma_{h}^{2}\right)\;\;\;\;\;\;\;\;\;\;\;\;\;\;\;\;\;\;\;\;}$$

Using (x,y) as the center coordinates of the character, and $\left(\sigma_{w},\sigma_{h}\right)$ as the variances of the Gaussian filter, which are proportional to the height and width of the bounding box corresponding to the character. If we denote the ratio between σ and each dimension of the bounding box as ω, they can be formulated as $\left(\sigma_{w}=\omega \times W\right)\;and\;\left(\sigma_{h}=\omega \times H\right)$. With overlaps in the Gaussian filters, the maximum values for the overlapping locations are taken into account, and we ignore the rest. To handle the imbalance issue for positive and negative examples, focal loss is adopted [22] for low-weight examples by assigning more weights to them. The designation of the center classification loss is as follows: (4)Lc−pred=−1S∑i=1Wn∑j=1HnδijFL(pijc)
where
(5)δij={1                if yijc=1(1−fijc)γ otherwise  ,           |fijc=Gijs=1,2,..,Smax(xs,ys,σws,σhs) 
(6)$$FL\left(p_{ij}^{c}\right)=\{\begin{array}{c} {\left(1-p_{ij}^{c}\right)}^{\gamma }log\left(p_{ij}^{c}\right)\;\;\;if\;\;y_{ij}^{c}=1\;\\ {\left(p_{ij}^{c}\right)}^{\gamma }log\left(1-p_{ij}^{c}\right)\;\;\;\;otherwise \end{array}$$

We let $p_{ij}^{c}$ be the confidence score, with a value of either 1 or 0, which determines if the location is the center or non-center of the character; $y_{ij}^{c}$ indicates ground truth annotations, and takes values between 0 and 1, where $y_{ij}^{c}=1$ when the location is the center of the character. $\delta_{ij}$ is a hyper-parameter that alleviates the effects of negatives surrounding the positives by providing low weights to those negatives. In this regard, the Gaussian filter $f_{ij}^{c}$ is employed to minimize the impacts of negatives on the total loss function by experimentally setting the hyper-parameter γ to 4 to control ambiguity.

Let $L_{c},\;R_{c},\;B_{c},\;U_{c}$ denote the four distances from center at the location $d_{e}\left(i,j\right)$ to the four corners of the bounding box, where
(7)$$L_{c}=d_{0}\left(i,j\right)=x-x_{0,\;\;\;\;\;}R_{c}=d_{1}\left(i,j\right)=x_{1}-x\;\;U_{c}=d_{2}\left(i,j\right)=y-y_{0,\;\;}\;\;\;B_{c}=d_{3}\left(i,j\right)=y_{1}-y\;\;$$

Here, $\left(x_{0,}\;y_{0}\right)$ and $\left(x_{1,}\;y_{1}\right)$ are the upper-left and bottom-right corners of the bounding box.

To compute the intersection over union IOU between the bounding boxes for ground truth and the predictions, we followed [23] to compensate for the losses of location information caused by moving away from the center. 

Then, we can write the whole regression loss as follows:(8)Lreg=1∑ℕ(d(i,j))∑(i,j)ℕ(d(i,j))LIOU(gr,pr),    |ℕ(d(i,j))={1   if de(i,j)>00         otherwise

The total loss function is the contribution of the three above losses and can be presented as
(9)$$L=L_{c-pred}+\emptyset_{1}L_{cl}+\emptyset_{2}L_{reg}$$

Experimentally, $\emptyset_{1}\;and\;\emptyset_{2}$ are set to 1 and 3, respectively. Furthermore, principal data augmentation methods are used to avoid overfitting; these methods apply random cropping patches from the input image, flipping them horizontally, and then a slight change in the color values is added by randomly using jittering color, which edits the brightness, saturation, and contrast.

## 4. Experiments

### 4.1. Datasets

Owing to the lack of old documents with Chinese characters, and cooperating with a team from Kyungpook National University (KNU), we collected and implemented a challenging scenario that contains separate handwritten characters in the Chinese language by scanning documents. The scanned data consisted of three groups with three different scripts: densely distributed of small-size characters, a few characters arranged vertically in a large, empty space, and containing very large-scale characters from cropped images. Moreover, images with only empty scripts were applied for augmentation purposes. More information about these datasets, along with the number of images in each group, is provided in Figure 4. The number of characters is different from one kind of image to another, from left to right in Figure 4, around 9, 54, 299, and 0, respectively.

### 4.2. Training Details

Using Python with PyTorch [24], the proposed model was implemented.

ResNet-52 [16], which is pretrained on Image-Net [17], was used as the backbone. To stay within the GPU memory limit, a batch size of one image for each GPU (GTX 1080Ti) was applied; four GPUs were employed in our experiments. Optimizing the model was done using the optimizer of Adam [25]. We set the learning rate to 1 × 10^−4^. The network was trained in an E2E manner until optimum performance was reached. Augmentation of basic styles, such as crop, color variation, and flipping, were used, and to improve the issue of positive-to-negative imbalance, On-line Hard Example Mining [26] was also applied. 

The proposed FC-MSCCD could detect Chinese characters effectively and accurately after the training phase, and the character score, which represents the accuracy of character prediction, was progressively boosted. Figure 5 depicts the character score map during the phase of training phase. At the beginning of the training phase, the character score was not high for new characters, but as training proceeded, the model learned the patterns of novel characters and finally found the character region accurately.

### 4.3. Ablation Study

#### 4.3.1. Reliability of the Center Heatmap

We suggest providing information about the character region using the center position lies because center-ness is a high-level feature position that can automatically predict the location of each character. This process of finding the center point proved the quality and efficiency of localization, whether the character is small or large. Table 1 compares other methods that use other feature points, such as top-left and bottom-right corners, as high-level features, as mentioned in [27]. After applying these methods to our model, we found that predicting one corner to find the character works worse than representing the character as a combination of two corners. An output heatmap with a pair of corners can provide good detection and good performance. However, it performed less well than using the center point by roughly 1.3%, with IOU = 0.5, and this gap is surprisingly greater for IOU = 0.7. This can be attributed to the quality of the center-ness, as expected when realizing the overall character details, which positively affected the training phase. We evaluated the model by employing a log-average miss rate (MR). However, instead of using this rate over False Positive Per Image (FPPI), False Positive Per Character (FPPC) was integrated for our experiments, in the range $\left[10^{-3},10^{-1}\right]$, indicated as MR/FPPC.

#### 4.3.2. Significance of the Regression to Four Corners 

For bounding box generation, finding the center leads to predicting the bounding box with respect to the four corners. In this case, regression to the four corners is the most essential component in order to produce the bounding box. In terms of generalization, we confirmed that the proposed model is capable of achieving better performance when learning new patterns of characters, in comparison with using only one or two corners to create the bounding box, which makes FC-MSCCD a kind of dynamic model that avoids overfitting issues. Table 2 shows the effectiveness of our model w.r.t four-corner regression.

#### 4.3.3. Reliability of Locations and Scale Variations

In order to cope with the enormous variety of character sizes, we use features from different levels, since fine details can be obtained from low-level features, while coarse information needs higher-level features. In this regard, a progressively combined feature technique is used. Furthermore, to enrich the resolution and provide more accurate details, blend connections between upsampling levels are enclosed. Using these two techniques of gradually combining features and applying the blend connections adds significant accuracy to character prediction and helps complete the training phase at a lower marginal cost. Table 3 compares network structures with and without blend connections. S1-1 is the feature map generated w.r.t the input image by downsampling *n* = 2, and is used here just for comparison purposes.

As noticed with blend connections, evaluation with IOU at 0.5 revealed that the best detection with the lowest miss rate was for the S2 feature map. We can clearly see that the blend connection provides notable performance for prediction over S4, with IOU at 0.5 and 0.7. The detector performed poorly for localization on this feature map without blend connections, and we can easily see that S2 without using blend connections provided the best prediction for IOU at 0.5. However, it produced poor detection at IOU = 0.7, in comparison with S1-1, which proves that the high-level feature map provided promising performance on the localization issue. After analyzing the importance of multi-level features with blend connections, we can say that the blend connection significantly improves detector performance since it can refine feature representation as it goes deeper but without extra computations.

#### 4.3.4. Reliability of the Feature Map Concatenation

As illustrated previously, combining features from different levels plays a significant role in improving detector performance because it enhances awareness of the character region using more information about character location. To that end, we conducted comparisons to investigate the optimal concatenation from these multi-level feature maps. Due to the powerful features represented by ResNet-52 [16], we employed it as a backbone.

The last combined feature maps are (Hn×Wn) w.r.t. the input image size (H×W). Here, n is set to 4, since a higher value gives inaccurate details about character location, whereas a smaller value adds to the processing time and leads to sub-optimal prediction. For computation cost considerations, we ignore the first feature map at n=2. As shown in Table 4, optimal detection is obtained by joining $\left\{d_{2},d_{3},\;d_{4}\;and\;d_{5}\right\}$ feature maps.

### 4.4. Comparisons with SOTAs and Investigations

Using ResnNet-52 layers, comparisons with SOTA methods were comprehensively made to evaluate our algorithm in terms of MR/FPPC using our database. Experiments were conducted on small-scale, medium-scale, large-scale, and multi-scale characters. Table 5 shows the results, and SOTA performance was achieved by the FC-MSCCD detector, which beats other detectors of Chinese character detection at different aspect ratios. Moreover, FC-MSCCD performed fairly well based on evaluations of the sub-datasets with small-, medium-, and large-scale characters (one subset for each training process).

Figure 6 presents the results of our method applied to our merged dataset. We used the predicted boxes overlaid on ground truths.

Figure 7 shows the detection results represented by blue boxes around the characters from our benchmark using three different models: FC-MSCCD, CCB-SSD, and HCCR-GoogLeNet. FC-MSCCD outperformed the other two algorithms in terms of accuracy.

Generally speaking, FC-MSCCD is a dynamic and simple detector based on the freedom of the default boxes to localize characters with high accuracy. In addition, center dots with regressions to four corners, which resulted from the annotations of the bounding boxes, are the only requirements for training the model, which means it is a light-weighted model requiring uncomplicated training. However, this technique may have difficulties when applied to generic text detection in other documents without some modifications to annotations or even some improvements in algorithms to suit other types of text or annotations.

When it comes to comparisons with studies based on default boxes, they impede training due to the large number of anchors with background information, which exhausts the process. To remedy this, FC-MSCCD handles it by searching for the center of each character instead of dealing with tedious anchors. As shown in Figure 8, our proposed model provides competitive performance against other models in terms of accuracy when using default boxes for training. On the other hand, employing heatmaps to find the centers of characters makes our model more robust than other detectors, even against the SSD competitor, as depicted in Figure 9.

Figure 10 shows the vitalization results of the Gaussian heatmaps.

## 5. Conclusions

A competitive Chinese character detector is presented in this work, named FC-MSCCD, which can detect multi-scale characters in an E2E fashion. We implement a new structure with a fusion of multi-level features and connections that blends the previous and current output from each up-sampling stage so that the coarse and fine representations can be considered together, and a higher resolution can be obtained. Furthermore, the novel standpoint of representing Chinese characters as centers facilitates the training phase and allows predictions that are free from complex and costly processes. Performance was evaluated under a challenging scenario of benchmarks consisting of old manuscripts of Chinese characters, which were collected in cooperation with a team from KNU. As an advanced step in the future, further explorations will focus on fine-tuning the abilities and generalizability of our FC-MSCCD detector to accommodate recursive character multi-language styles.

## Figures and Tables

**Figure 1 sensors-23-02305-f001:**
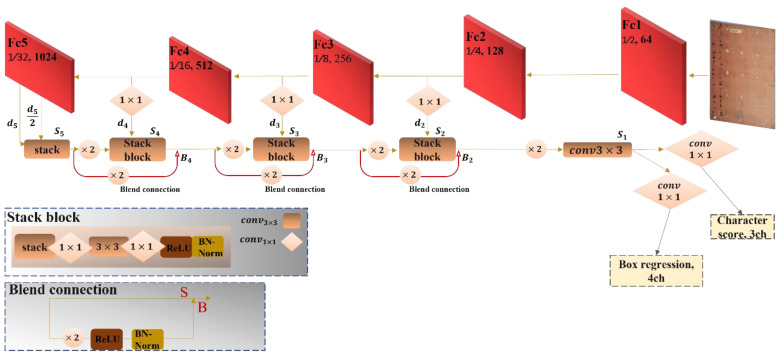
Overall design of the proposed FC-MSCCD. It has two parts: feature extraction and detection head. Feature extraction embraces two branches: trunk and stack branches with blend connections, which is residual-style learning. The detection head mainly consists of multi-channels for center prediction and bounding box regression after merely adding a 3×3 Conv layer followed by a 1×1 Conv layer. The schematics in the lower-left corner explain the stack block components and the flow of the blend connection.

**Figure 2 sensors-23-02305-f002:**
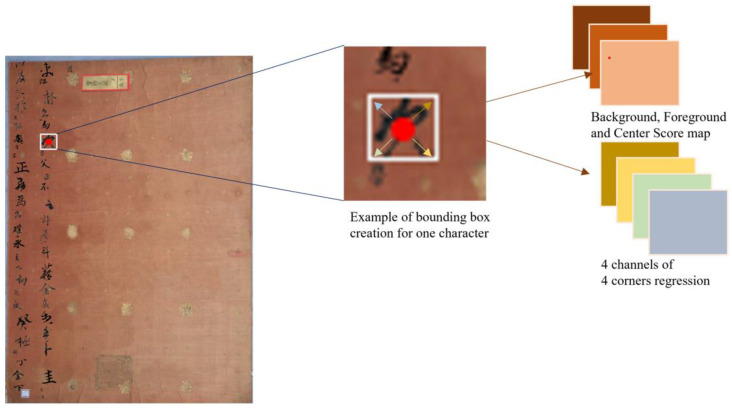
The process of automatically generating the bounding box from the center is referenced as a principle to regress to the four corners. This is based on a multi-channel procedure in which one channel is a heatmap finding the center of the character (red dot), two channels are for background and foreground classification, and four channels are for regression to the four corners of the bounding box.

**Figure 3 sensors-23-02305-f003:**
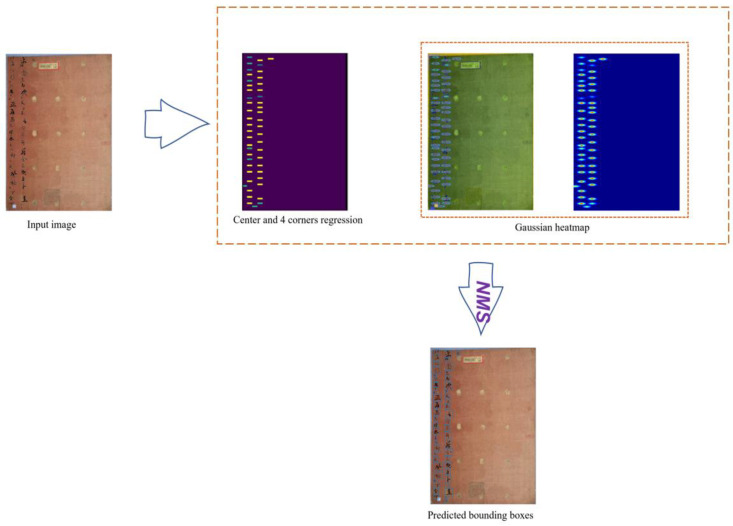
Illustration of automatically finding character centers and building bounding boxes from bounding box annotations, clarifying positives and negatives. From left to right, the input image already has box annotations that are generally obtained by algorithms using default boxes for predictions. Then, a center heatmap with regression to the four bounding box corners indicates all centers as positives. Otherwise, they are negative. Finally, to eliminate the fuzziness of these negatives fencing the positives, Gaussian normalization, presented in Equation (2), is presented in the third and fourth images on the right. The output prediction is adopted after Non-Maximum Suppression (NMS).

**Figure 4 sensors-23-02305-f004:**
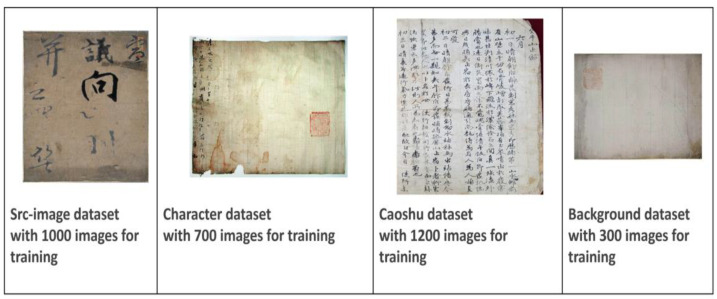
Samples of the four benchmark sub-sets. From left to right, the Src-image dataset, the Character dataset, the Caoshu dataset, and the Background dataset note the number of images used from each of them for training.

**Figure 5 sensors-23-02305-f005:**
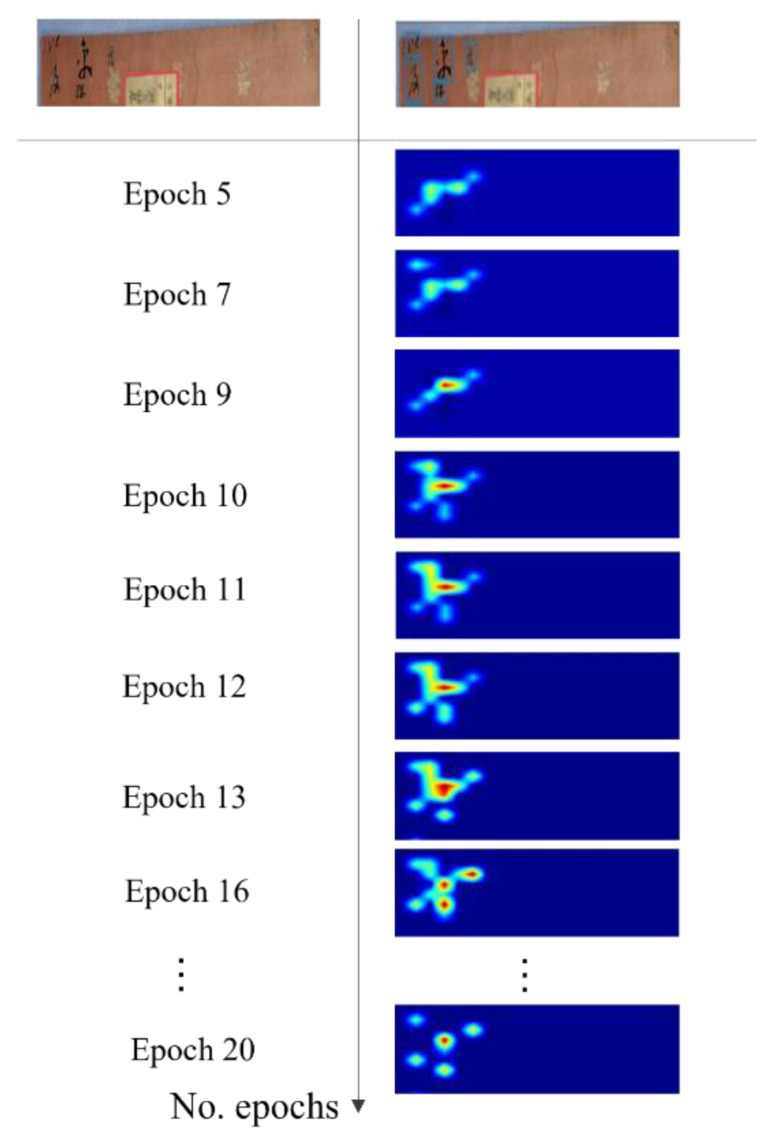
The training phase, visualizing character score maps.

**Figure 6 sensors-23-02305-f006:**
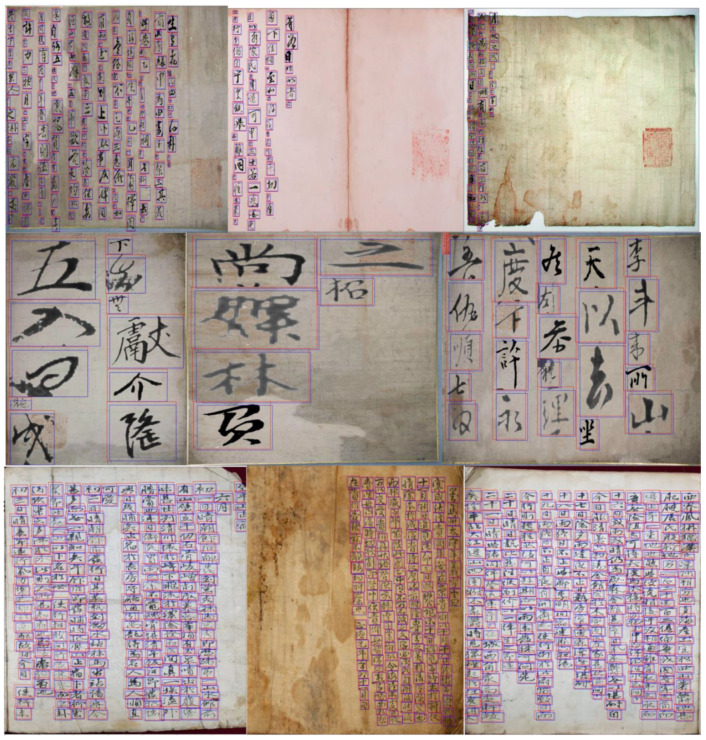
Detection examples on our benchmark using a merged dataset for generalization purposes. Our model is capable of finding multi-scale Chinese characters in historical manuscripts, and achieved extremely accurate results. Even it proved to have the ability to recognize backgrounds without detection errors.

**Figure 7 sensors-23-02305-f007:**
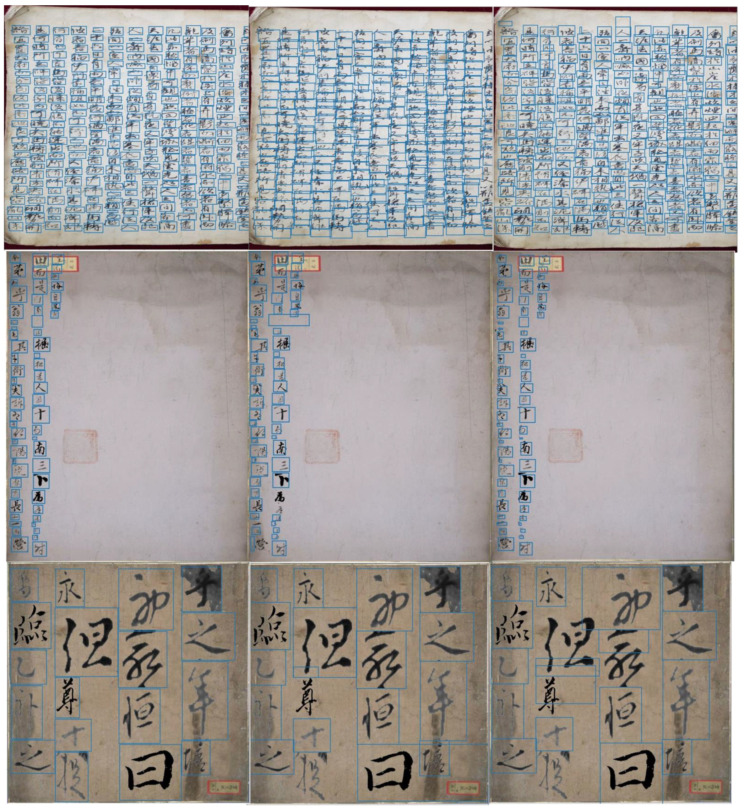
Visualization comparisons of three different models. From the left column to the right column, FC-MSCCD, CCB-SSD, and A human-inspired recognition system. In our benchmark, the first row shows detection results for the Caoshu dataset, the second row shows detection results for the Character dataset, and the last row shows detection results for the Src-image dataset. We used a merged dataset in our experiments.

**Figure 8 sensors-23-02305-f008:**
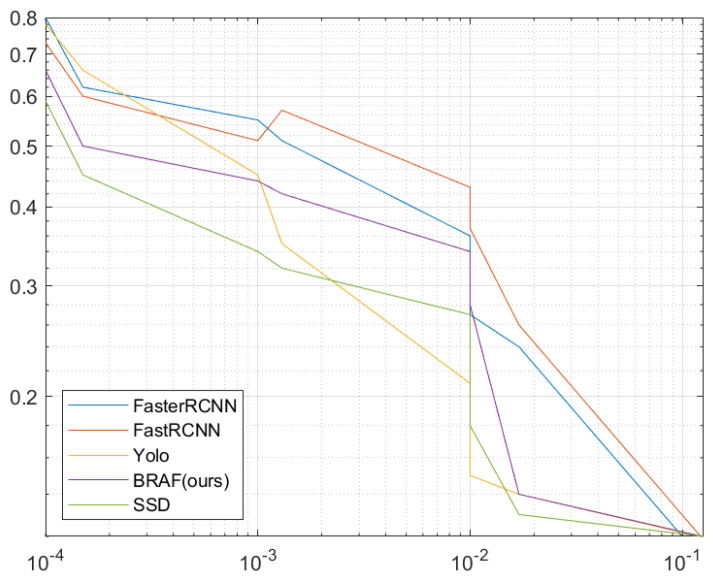
Missing Rate (MR) versus False Positive Per Character (FPPC) for training with default boxes for comparison of our FC-MSCCD model with other SOTA models.

**Figure 9 sensors-23-02305-f009:**
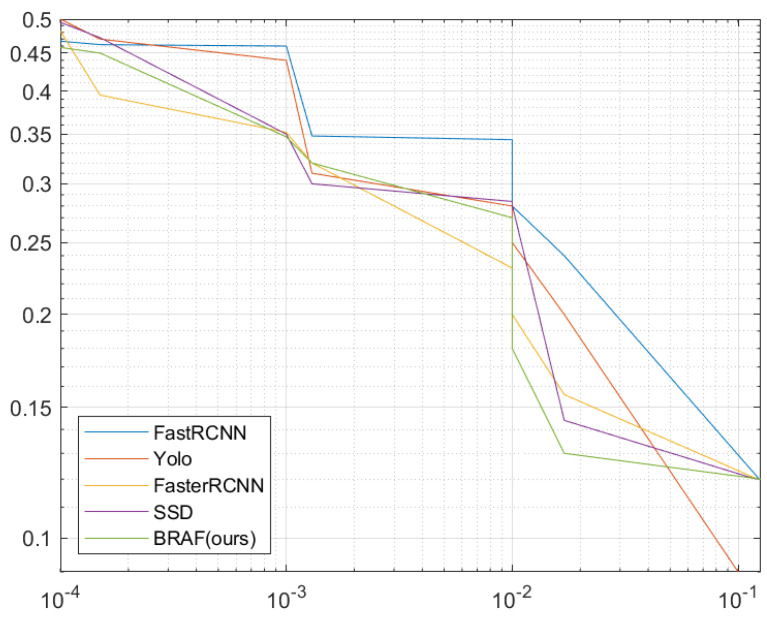
Missing Rate (MR) versus False Positive Per Character (FPPC) for finding the centers of characters in heatmaps for comparison of our FC-MSCCD model with other SOTA models.

**Figure 10 sensors-23-02305-f010:**
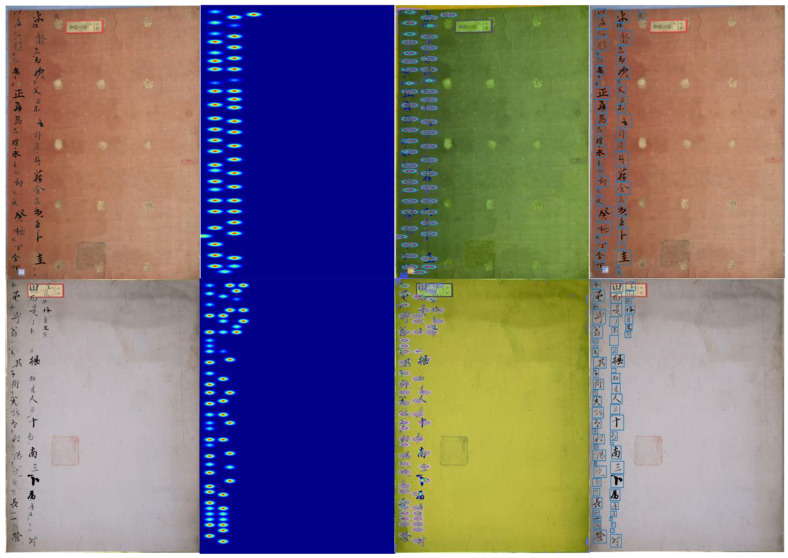
Visual results of a Gaussian heatmap. The first column is the input image, the second column is the Gaussian heatmap, the third column is the characters overlaid on the Gaussian heatmaps, and the last column is the predicted bounding boxes.

**Table 1 sensors-23-02305-t001:** Comparisons of finding different points each time as a high-level semantic feature.

Position IOU	MR/FPPC (%)	
	0.5	0.7
Center	4.56	35.44
Pair of corners	6.86	29.61
Left-top corner	7.90	42.52
Right-bottom corner	8.23	45.22

**Table 2 sensors-23-02305-t002:** Comparisons of different regression methods with respect to bounding box corners.

Regression Technique IOU	MR/FPPC (%)	
	0.5	0.7
Four corners	4.56	35.44
Two corners	5.98	42.61
One corner	6.99	46.73

**Table 3 sensors-23-02305-t003:** Performance comparison using the down-sampling branch with and without blend connections.

Feature Maps for Prediction IOU	Blend Connections	MR/FPPC (%)	
		0.5	0.7
S1-1		5.65	32.09
S2		5.02	37.71
	◯	4.82	29.13
S3		7.44	56.53
	◯	6.48	34.91
S4		21.34	77.56
	◯	8.34	35.74

**Table 4 sensors-23-02305-t004:** Performance of various feature map concatenations for multi-scale purposes.

Feature Maps	No. Parameters MB	MR/FPPC (%)	Test Time (ms/Image)
$d_{2}\;d_{3}$	10.7	11.12	40.2
$d_{3}\;d_{4}$	22.8	6.02	45.3
$d_{4}d_{5}$	40.4	6.01	52.5
$d_{2}\;d_{3}\;d_{4}$	23.1	7.43	50.0
$d_{3}\;d_{4}\;d_{5}$	46.0	4.93	60.1
$d_{2}\;d_{3}\;d_{4}\;d_{5}$	45.6	4.82	62.2

**Table 5 sensors-23-02305-t005:** The FC-MSCCD model beat other SOTA models on our benchmark dataset in terms of multi-scale detection and even with different scales in the sub-dataset.

Algorithm	Backbone	Small-Scale MR/FPPC (%)	Medium-Scale MR/FPPC (%)	Large-Scale MR/FPPC (%)	Multi-Scale MR/FPPC (%)
A human-inspired recognition system [28]	DenseNet	6.31	6.23	5.35	-
HCCR-CNN12layer [29]	LeNet	-	5.62	5.35	-
GWOAP [30]	CNN	9.05	6.20	6.09	-
FAN-MCCD [31]	ResNet-52	4.90	4.71	4.98	4.95
Two-stage hierarchical deep CNN [32]	CNN	-	4.99	5.01	-
CCB-SSD [33]	ResNet-34	7.38	6.94	6.82	6.70
Recognition of Japanese Connected Cursive Characters [34]	CNN	7.75	7.56	6.91	-
FC-MSCCD (ours)	ResNet-52	4.51	4.63	4.82	4.82

## Data Availability

The Chinese figures used in the manuscript were collected and created by cooperation between the Yeungnam University team and the Kyungpook National University team. The permission of the other team was obtained with no need for copyright since it is our own dataset. The database is available online (http://dila.co.kr/index.php, accessed on 28 October 2021).

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
