# Peer review of "Handwritten Multi-Scale Chinese Character Detector with Blended Region Attention Features and Light-Weighted Learning"

_sensors, 2023, doi:10.3390/s23042305_

Round 1

Reviewer 1 Report

The authors presented "Handwritten multi-scale Chinese character detector with blended region attention features and light-weighted learning". The paper is interesting for both researchers and practitioners. Experiments are well explained. Results are clear and concise. 

However, I have some concerns that must be improved in the revised version before acceptance. 

Major Issues:

Q1: Abstract

It is recommended to add the challenge in the summary section.

Q2: Introduction

It is recommended that the authors detail the motivation for the innovation of the proposed approach in the introduction section.

Q3: Introduction

It is recommended that an overview of the remaining section of the article be added at the end of the Introduction section.

Q4: Existing Works 

It is suggested that the authors add in this sections the innovation of doing the proposed method by themselves compared with the existing method.

Q5: Conclusion

It is recommended that the sections detail the shortcomings of the work done and provide guidance for future work.

Reviewer 2 Report

1.In the introduction part, it is mentioned that the current deep learning methods have the phenomenon of excessive segmentation, but the description of the proposed network seems to solve the problem with high time cost. Please explain it if possible

2.The proposed method adopts the UNet structure, so it is suggested to briefly introduce the UNet network and its applications in part Feature extraction with a feature fusion model of the related work, the following papers may be helpful:

Swin-Unet: Unet-like Pure Transformer for Medical Image Segmentation. arXiv, 2021

A Change Detection Method Based on Multi-Scale Adaptive Convolution Kernel Network and Multimodal Conditional Random Field for Multi-Temporal Multispectral Images. Remote Sensing, 2022

ICA-UNet: ICA Inspired Statistical UNet for Real-time 3D Cardiac Cine MRI Segmentation. MICCAI 2020, 2020

3.There are spelling errors in lines 127 and 369. It is suggested to check the manuscript carefully.

4.In the network architecture part of the methodology, the size of Si  and Bi can be marked out.

5.In line 219, two hyperparameters are mentioned. Please give the details about how to select the value of the hyperparameters?

6. The sixth and seventh references are repeated, please check.

Round 2

Reviewer 2 Report

No more comments